# Regenerative Strategies in Cleft Palate: An Umbrella Review

**DOI:** 10.3390/bioengineering8060076

**Published:** 2021-06-03

**Authors:** Inês Francisco, Anabela Baptista Paula, Bárbara Oliveiros, Maria Helena Fernandes, Eunice Carrilho, Carlos Miguel Marto, Francisco Vale

**Affiliations:** 1Institute of Orthodontics, Faculty of Medicine, University of Coimbra, 3004-531 Coimbra, Portugal; anabelabppaula@sapo.pt (A.B.P.); fvale@fmed.uc.pt (F.V.); 2Institute of Integrated Clinical Practice, Faculty of Medicine, University of Coimbra, 3004-531 Coimbra, Portugal; eunicecarrilho@gmail.com (E.C.); cmiguel.marto@uc.pt (C.M.M.); 3Coimbra Institute for Clinical and Biomedical Research (iCBR), Area of Environment Genetics and Oncobiology (CIMAGO), Faculty of Medicine, University of Coimbra, 3004-531 Coimbra, Portugal; boliveiros@fmed.uc.pt; 4Centre for Innovative Biomedicine and Biotechnology (CIBB), University of Coimbra, 3004-531Coimbra, Portugal; 5Clinical Academic Center of Coimbra (CACC), 3004-531 Coimbra, Portugal; 6Laboratory of Biostatistics and Medical Informatics (LBIM), Faculty of Medicine, University of Coimbra, 3004-531 Coimbra, Portugal; 7LAQV/REQUIMTE, Faculty of Dental Medicine, University of Porto, 4200-393 Porto, Portugal; mhfernandes@fmd.up.pt; 8Institute of Experimental Pathology, Faculty of Medicine, University of Coimbra, 3004-531 Coimbra, Portugal

**Keywords:** autogenous graft, biomaterials, cleft palate, bone grafting, regenerative

## Abstract

(1) Background: Alveolar bone defects or decreased alveolar bone height and width may have different causes, such as cleft palate. Regenerative procedures in oro-dental defects are challenging due to anatomical factors and the distinct cell populations involved. The iliac crest bone graft remains the gold-standard for cleft palate closure. However, tissue regeneration approaches have been employed and their outcome reviewed, but no conclusions have been made about which one is the gold-standard. (2) Methods: this umbrella review aims to critically appraise the effectiveness of the current approaches in bone defects regeneration in non-syndromic patients with cleft palate. A search was performed in PubMed, Cochrane Library, Scopus, Web of Science and EMBASE databases. (3) Results: Systematic reviews of randomized and non-randomized controlled trials with or without meta-analysis were included. Nine articles were included in the qualitative analysis and five in the quantitative one. The included studies quality was evaluated with AMSTAR2. (4) Conclusions: The use of new regenerative strategies, such as bone morphogenic protein 2, appears to provide similar results regarding bone volume, filling, and height to the standard technique with the iliac crest bone graft.

## 1. Introduction

Cleft lip and/or palate (CLP) is a craniofacial malformation with a prevalence in newborns of 14 per 10,000 live births worldwide [1]. Although not yet fully understood, several genetic and environmental risk factors are associated with CLP [2]. Smoking, gestational diabetes, and genetic association with the IRF6, VAX1, and PAX7 genes are consistently reported [3,4]. Most CLP patients have several hearing, feeding, speaking, and dentofacial development complications, leading to a long-term impact on the patient’s facial anatomy and self-esteem [5,6]. Additionally, new evidence suggests a common basis for orofacial cleft and cortical interneuronopathy, supported by cellular and molecular central nervous system (CNS) alterations in these patients [7].

CLP treatment requires a coordinated interdisciplinary team that considers the patient’s and family’s needs [8,9]. One of the endpoints in CLP care is a secondary bone graft, which presents several benefits such as support of unerupted teeth that will erupt into the bone graft, support of alar bases (promoting nasal and lip symmetry), closure of oronasal fistulas and cleft maxillary segments stabilization [10,11,12]. The bone graft is usually performed when the canine has one-half to two-thirds root formed. However, a recent systematic review (SR) that compares bone graft performed early at approximately 5–6 years and at the conventional time at 9–11 years did not find significant differences between the two protocols [13]. Since its introduction by Boyne and Sands in 1972, autologous bone graft using cancellous bone remains the gold-standard until today [14]. Cancellous bone grafts have abundant osteogenic surface cells, which allow new bone formation [15]. The origin of the autogenous bone graft depends on numerous factors, namely, the surgeon’s experience, the volume of alveolar defect, and the morbidity of the harvest area [16]. Typically, an autologous bone graft from the iliac crest (ICBG) is the conventional donor site [17]. However, others with less morbidity and lower bone resorption rate have been suggested, such as cranial bone, mandibular symphysis and tibia [18,19,20]. Nevertheless, autologous bone graft has several limitations, such as limited donor supply and/or self-renewal capacity, operative time, costs and donor site morbidity [21]. Furthermore, it is also reported that bone resorption can be approximately 40% after one year of bone graft, which may increase the need for reintervention [18].

Therefore, tissue regeneration presents as a new emerging and alternative approach to conventional bone grafts in cleft patients. Besides the bone regeneration capacity, these strategies can modulate inflammation and enhance the healing process [22]. Many substitute materials or agents alone or combined with autogenous bone have been suggested to regenerate bone, such as growth factors like bone morphogenic protein 2 (BMP-2) platelet-rich fibrin, bone scaffolds with or without cell treatment (e.g., mesenchymal stem cells (MSCs) or osteoblast), biocomposites (e.g., calcium phosphate and hydroxyapatite) and haemostatic agents (e.g., fibrin glue) [16,23,24,25,26,27,28,29]. Stem cell-based therapies have been explored based on several stem cell types: bone-marrow stem cells, adipose-derived stem cells, umbilical cord mesenchymal stem cells, and others [30,31]. Besides the direct effect on bone regeneration, the activation of resident stem cells can present a broader impact, for instance, on the CNS alterations previously described [7]. For this specific purpose, adult neural stem cells (NSCs) are the main cell population involved. These cells are maintained throughout life on the lateral ventricles and the hippocampus, have a multipotent capacity and long-term stemness, and can produce neurons and microglia cells [32]. In adults, they are mainly in a quiescent state, characterized by an increased expression of Sox9 and Id2, Id3, Id4, Vcam, Cdh2, Klf9 and Lrig1 [33]. When stimulated, NSCs can become active and contribute to tissue regeneration [34]. It must be pointed out that evidence regarding some of these therapeutic methods is often weak due to the lack of standardization across studies.

Despite a recently renewed focus on bone graft materials, an overall synthesis and appraisal of these reviews is still lacking [35]. To clarify this, we conducted an umbrella review to assess the clinical effectiveness of regenerative strategies on the treatment of secondary bone graft in cleft patients, which would be advantageous for readership since it synthesizes what we know in a single paper. Therefore, this umbrella review aims to answer the following focused question: “Is the regenerative capacity of new bone graft strategies more effective than the gold-standard?”

## 2. Materials and Methods

### 2.1. Protocol Registration

This review was registered in the International prospective register of systematic reviews (PROSPERO) with provisional number 240534. This study was conducted according to the Cochrane and PRISMA guidelines for systematic reviews [36,37,38].

### 2.2. Review Question

The purpose of this umbrella review was to assess the clinical effectiveness of regenerative strategies on cleft patients treatment, with the following PICO question:Population—cleft palate patients (non-syndromic cleft lip and/or palate patients (unilateral and bilateral) of all ages that underwent regenerative strategies as part of their treatment);Intervention—undergoing treatment approaches with regenerative strategies (all available treatment approaches for cleft palate closure: conventional autologous graft from different origins, alloplastic material, platelet-rich fibrin, platelet-rich plasma, resorbable collagen sponge, bovine-derived hydroxyapatite, allogeneic bone material, demineralized bone matrix, acellular dermal matrix and human bone morphogenetic protein 2);Comparison—different available regenerative strategies;Outcome—bone regeneration.

### 2.3. Eligibility Criteria

The included studies were all systematic reviews of randomized trials, non-randomized controlled trials and case control studies. Studies that included regenerative strategies but evaluated other outcomes, such as quality of life, feeding problems, phonetics problems, assessment method, and velopharyngeal function, were excluded. Case reports, case series and literature reviews were also excluded.

### 2.4. Search Strategy

A standardized literature search was performed in electronic bibliographic databases (MEDLINE via Pubmed, Web of Science databases, Cochrane Library, Scopus and EMBASE), up to 27 February 2020. The search strategy can be seen in Table 1.

The search for unpublished articles in the grey literature was carried out through the websites Proquest (https://www.proquest.com (accessed on 24 February 2021) and OpenGrey Europe (https://opengrey.eu (accessed on 24 February 2021).

The reference lists of the relevant articles were manually searched to explore additional studies.

### 2.5. Study Selection and Data Collection

The search and study selection were carried out by two reviewers (IF and ABP). If the two reviewers could not agree on a certain study’s eligibility, another reviewer resolved disagreements (CMM). The articles were screened based on the titles and abstracts according to the eligibility criteria by the two independent reviewers mentioned above, in duplicate. Subsequently, full texts were screened for potential inclusion and disagreements were resolved through mediation with the third reviewer.

From each included study, the authors extracted the following information: authors and publication data, design of the included studies and their number, sample characteristics (age of participants, intervention and comparative unit), primary outcome, and evaluated parameters in the study. If present, the meta-analysis model used, effect size with a 95% confidence interval, I^2^ statistic, heterogeneity, and GRADE evidence were extracted.

A descriptive summary of each study’s main findings was performed.

### 2.6. Quality Assessment

The selected studies qualitative assessment was performed using the Assessment of Multiple Systematic Reviews (AMSTAR 2) (https://amstar.ca/mascripts/Calc_Checklist.php (accessed on 8 March 2021)) checklists. AMSTAR2 checklists contain several questions directed only to systematic reviews under evaluation [39,40]. Two reviewers (IF and ABP) performed the quality assessment of the studies in duplicate and independently, categorizing them as: high quality if none or only one of the parameters is weak; moderate quality if more than one parameter is weak; and poor quality if there are several weak parameters or a major failure. Three other reviewers (CMM, EC and FV) also independently assessed the studies’ quality, and, if in disagreement with the initial evaluation, this point was discussed together.

### 2.7. Statistical Analysis

A random effects meta-analysis over the standardized mean difference between ICBG and BMP-2 was conducted whenever at least three systematic reviews were reporting the same synthesis measure. A synthetic standardized effect size was computed using the total number of subjects included in the previous meta-analysis studies, and its correspondent 95% confidence intervals and *p*-value for the comparison between ICBG and BMP-2 along with forests plots and funnel plots. The Egger regression and Begg–Mazumdar test were applied to assess publication bias regarding the dependency of outcome measures from previous meta-analysis variability. Publication bias was also evaluated for the three meta-analyses performed.

The analysis was performed through the ‘metafor’ package of R, version 4.0.3 implemented in R Studio, version 1.3.1093 and was analyzed at a 5% significance level.

## 3. Results

### 3.1. Study Selection

The flow chart for this umbrella review can be seen in Figure 1. The search in the different databases resulted in 1317 articles, with no paper retrieved from grey literature or from the manual search. After the articles were screened by title and abstract, 20 full-text articles were assessed for eligibility. Full-text screening led to the exclusion of 11 articles due to several reasons: one case-report, two literature reviews, four with different interventions or outcomes, four with other measures of bone regeneration and one poster abstract. Nine systematic reviews were included in the qualitative analysis and five in the quantitative analysis (meta-analysis).

### 3.2. Characteristics of the Included Reviews

The characteristics and results of the included reviews are presented in Table 2.

This umbrella review includes nine systematic reviews (SR) comprising 56 RCTs, 61 non-RCTs and nine case-control cases. Five SRs were registered in PROSPERO and four did not report any registration. Most SRs used the Cochrane (4) ROB tool, the others used MOOSE/Strobe (1), Oxford Centre for Evidence-Based Medicine (1) or Newcastle-Ottawa (1), and two studies did not report any assessment. The evidence quality was considered high in 15, moderate in 11 and low in 39 of the studies included in the SRs. A SR that used the Newcastle–Ottawa tool was classified as level 6 and 7.

The included patients ranged in age from over five years old to adolescence (16 years old). However, in four reviews the age parameter was not reported.

In all reviews, the comparison or control technique used was the ICBG, which is considered the gold-standard. The interventions or new bone regeneration strategies were diverse, although BMP-2 was used in most studies. The other strategies were: bone substitutes (Bioglass, β-tricalcium phosphate, hydroxyapatite); cells (osteoblasts); other biomaterials (platelet rich plasma (PRP), demineralized dentinal matrix (DDM), demineralized bone matrix (DBM), acellular dermal matrix (ADM), Bio-guide, Membrane, Periosteum, deproteinized bovine bone (DBB)) with ICBG; or autogenous bone grafts from other sources (cranium, rib, tibia, mandible, symphysis and calvarium grafts).

The primary outcome evaluated was the volume, filling, height, rate and density of bone regeneration. The secondary outcomes evaluated were the failure rate, the postoperative infection, the persistence rate of the fistula and the length of hospital stay.

### 3.3. Quality of the Included Reviews

The quality of the included reviews evaluated by the AMSTAR2 tool can be seen in Table 3. All reviews presented the PICO question except one [41]. Successful registration was always carried out except in one revision [41] and in another in a partial form [16]. The inclusion criteria were omitted in some reviews since the PICO issue was already explained [16,42,43,44]. The search was only fully explained in two reviews [24,43]. However, data selection and extraction were carried out in duplicate, except in one review [16]. The list of excluded studies is not presented in a single review [16]. The description of the included studies not made in just one study [41]. In a review, the risk of bias assessment was not performed [45]. The funding of included studies was not reported in most of the reviews [16,24,42,45,46,47]. Three reviews did not present a meta-analysis and consequently do not have combined statistical results [16,41,44], ROB effects on the statistical combination [16,41,44,45] and ROB in the discussion [16,41,44,45,47]. Heterogeneity is discussed in some studies [24,41,42,43,44,47]. Publication bias was presented in only one review [43]. The study funding and the conflict of interests report were not referred to in one review [16]. Thus, with the application of the AMSTAR 2 tool criteria, four reviews were considered to be of low quality [16,41,44,45], four were considered to be of moderate quality [24,42,46,47] and one was considered to be of high quality [43].

### 3.4. Synthesis of the Best Evidence

#### 3.4.1. Bone Formation Volume Analysis

The quantitative analysis of previous meta-analysis synthetic measures was possible using three systematic reviews (Rosa et al., 2019, Uribe et al., 2019 and Xiao et al., 2020) [42,46,47] and showed no heterogeneity between previous synthetic measures (I^2^ = 0.00%). Besides this, the random effects model was still applied due to the number of studies in the analysis, and we found that the global standardized mean difference between ICBG and BMP-2 was not statistically significant (*p* = 0.704) and was estimated to be 0.08 mm^3^ ± 0.22 mm^3^ (95% CI: −0.35 to 0.51 mm^3^), as presented in Figure 2.

#### 3.4.2. Bone Formation Percentage Analysis

The quantitative analysis of previous meta-analysis synthetic measures was possible considering four systematic reviews (Rosa et al., 2019, Wu et al., 2017, Xiao et al., 2020 and Uribe et al., 2019) [42,45,46,47] and showed high heterogeneity between studies (I^2^ = 93.8%), perhaps due to Wu et al., 2017 [45], which is the only one presenting a statistically significant effect. Despite this, a global standardized mean difference between ICBG and BMP-2 was estimated and found not to be statistically significant (*p* = 0.184) and was estimated to be 67.92% ± 51.05% (95% CI: −32.13 to 167.97%), as presented in Figure 3.

#### 3.4.3. Bone Height

The quantitative analysis of bone height was possible with three systematic reviews (Uribe et al., 2019, Xiao et al., 2020 and Scalzone et al., 2019) [42,43,47] and heterogeneity between studies was found to be almost the maximum (I^2^ = 99.88%), as Xiao et al., 2020 [42] presented a large statistically significant effect when compared to the others. Despite this, a global standardized mean difference between ICBG and BMP-2 was estimated and found not to be statistically significant (*p* = 0.520) and was estimated to be 5.13% ± 9.97% (95% CI: −10.49 to 20.74%), as presented in Figure 4.

### 3.5. Quality of the Evidence

#### 3.5.1. Bone Formation Volume Analysis

The effect size reported by the studies used in the previous meta-analysis did not depend on the precision of the studies (Egger regression b = 0.04; *p* = 0.931), and was not correlated to the studies variance (Begg–Mazumdar: t_Kendal_ = 0.333; *p* = 0.999) indicating lack of publication bias concerning the influence of sample size and variability on the mean difference between treatments.

#### 3.5.2. Bone Formation Percentage Analysis

The effect size reported by the studies used in the previous meta-analysis did not depend on the precision of the studies (Egger regression b = −0.08; *p* = 0.939), and was not correlated to its variance (Begg–Mazumdar test: t_Kendal_ = 0.000; *p* = 1.000).

#### 3.5.3. Bone Height

The effect size reported by the studies used in the previous meta-analysis did not depend on the precision of the studies (Egger regression b = −4.17; *p* = 0.485), and was not related to the variance of the effects reported (Begg–Mazumdar test: t_Kendal_ = 0.333; *p* = 0.999), indicating lack of publication bias.

## 4. Discussion

This umbrella review aimed to synthesize the current literature regarding bone strategies in cleft patients, evaluating their success or failure based on systematic reviews with/without meta-analyses.

Conventionally, an autologous bone graft using cancellous bone is considered the gold-standard technique for bone graft [14]. All included studies that had a comparator group used autologous bone graft as a control. The effectiveness of tissue regeneration approaches was also investigated, with human bone morphogenetic protein being the most reported in 6 of the 9 systematic reviews included. BMP-2 is usually delivered in an alloplastic bone graft or scaffold and is an effective inducer of bone and cartilage formation, belonging to the transforming growth factor-beta proteins superfamily. This protocol avoids the limitations of autologous bone grafts, such as limited donor supply and donor site morbidity, and reduces the patient’s surgical stress, which may be related to the lower operative time and hospital stay length, reported by some studies [48,49]. However, some adverse effects such as nasal stenosis and localized graft-site oedema still persist [42,43,50].

Although some of the systematic reviews included indicated that BMP-2 treatment may benefit bone formation compared to autogenous bone graft, this umbrella review reported no significant difference between these two protocols regarding volume, filling, and bone height [43,47]. However, these findings should be interpreted with caution because clinical and methodological heterogeneity can influence the magnitude of the statistical heterogeneity reported. This umbrella review identified different heterogeneity factors, namely, the number of participants, type of cleft, the timing of outcome, and intervention design. Thus, the results of the included studies can be affected by this heterogeneity, and, consequently, the results of this umbrella review can also be affected. Additionally, some SR results could not be used for the meta-analysis due to different synthesis measures, decreasing the number of included studies, which can affect the obtained result. Nevertheless, the quality assessment revealed that only one study included has a high risk of bias, which gives more confidence in the results of this umbrella review. Therefore, both methods can be an option in cleft bone graft treatment.

The timing for the bone graft is a variable that clinicians should consider because it can influence the bone graft prognosis, endangering the support provided for teeth eruption, the continuity of dental arch and the closure of the oronasal fistula [50]. Despite the difference in the selected studies’ age groups, the majority performed the bone graft after 8 years of age, which is by the ideal timing reported in the literature [13,51,52]. Furthermore, a recent study by Brudnicki et al. reported that bone grafts performed before 8 years old might have a limited negative effect on craniofacial morphology [50].

Regarding the postoperative newly formed bone evaluation, most included studies used computed tomography. Even though this method has several advantages when compared with two-dimensional tools (e.g., control teeth eruption process into the bone graft and the assessment of the dimensional location of the bone graft), it has some limitations when compared with cone-beam computed tomography (CBCT) [53,54,55,56]. CBCT scans have a lower radiation dose, minimal scanning time (10–70 s) and allow the clinicians to scan a small region for specific diagnosis with less image artefact [57]. The orofacial cleft patients require a 3D analysis for the correct diagnosis since they present with several medical conditions, namely bone graft interventions, impacted teeth or supernumerary teeth. This is the reason why CBCT is indicated in orofacial cleft patients by the European Academy of Dental and Maxillofacial Radiology [58]. Therefore, further studies should use CBCT as an assessment tool to measure newly formed bone.

Follow-up times are not consensual, even though most of the trials included in the systematic review reported a follow-up of six months. This period is ideal for carrying out radiograph control since the remodeling process with cortical maturation occurs after six months, remaining stable until the 24th month. Regarding the included studies, only Scalzone et al. considered trials with at least six months of follow-up [43]. The remaining systematic reviews’ findings should be carefully analyzed since the remodeling process may not be completed.

This umbrella review has several strengths. Firstly, it provides a comprehensive overview of the available systematic reviews published following a registered protocol with transparent methodology. Moreover, the quality of the individual studies included was assessed using an AMSTAR 2 tool.

The findings of this umbrella review could be affected by the methodological and clinical heterogeneity among the included studies. Considering the risk of bias evaluated using the AMSTAR-2 tool, the items that presented more studies with low quality were: funding of included studies, discussion for the heterogeneity, ROB in the discussion, inclusion criteria and comprehensive search. In future systematic reviews, the authors should state their funding sources, namely by industry, as this may introduce a bias in the results presented. Most of the included studies failed to discuss the heterogeneity and the risk of bias, especially in justifying the inclusion of studies with different methodologies and the consequent bias. Therefore, their conclusions must be interpreted with caution. Before starting the systematic review, it is essential to reduce methodological flaws, bias and duplication risk. Several SRs included in this umbrella review presented some gaps in the search description and inclusion criteria definition, which potentially increase the risk of publication and reduce the comprehensive nature of the review. Thus, the registration of the protocol is recommended and will enhance the robustness in further research. Although systematic reviews are considered the most reliable evidence, the studies included in each SR also had associated bias. The methodological heterogeneity includes differences in the trial settings, missing a priori adequate sample size calculation, type of sample included (e.g., type of cleft, age groups), intervention protocols, bone measurement tools and follow-up times. Other variables may affect the analysis of primary outcomes since they affect bone remodeling, namely, the position of teeth on the bone graft, the cleft defect’s width, and the volume of grafted bone. The primary outcomes may also be affected by the clinician’s expertise and the research group scientific proficiency. Secondarily, most selected studies were categorized as having a low or moderate overall quality, which may decrease the findings’ certainty. Moreover, the included studies can perform an overestimation of the findings’ effects due to the inclusion of several publications of a single study or by excluding studies in other languages. Finally, SR without meta-analysis may present a high risk of bias. Therefore, the findings of this umbrella review should consider these limitations in the interpretation of results.

Future research should carry out blinded RCTs to control possible sources of bias such as randomization procedure, measurement tools and follow-up times. Moreover, the cost^–^benefit analysis of these new regenerative strategies in the bone graft is recommended since it plays a crucial role in healthcare systems’ decision-making.

## 5. Conclusions

This umbrella review suggests that BMP-2 and autogenous bone graft are both valid options since no differences were observed regarding volume, filling, and bone height in oral cleft patients’ surgery.

However, the findings should be analyzed cautiously due to several research gaps concerning the original studies’ methodological quality.

## Figures and Tables

**Figure 1 bioengineering-08-00076-f001:**
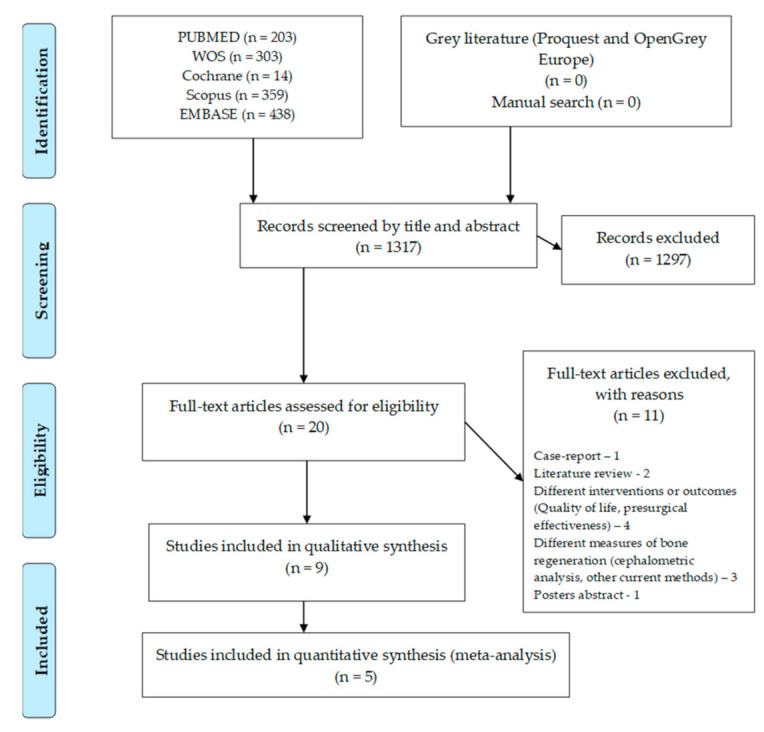
Flow chart for included systematic reviews.

**Figure 2 bioengineering-08-00076-f002:**
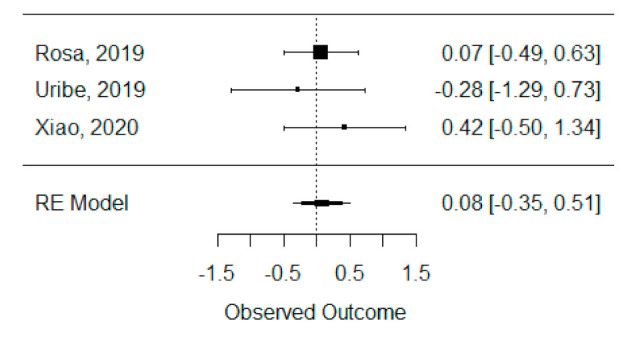
Forest plot for the standardized mean difference obtained for bone formation volume analysis synthetic measures reported in the included systematic reviews.

**Figure 3 bioengineering-08-00076-f003:**
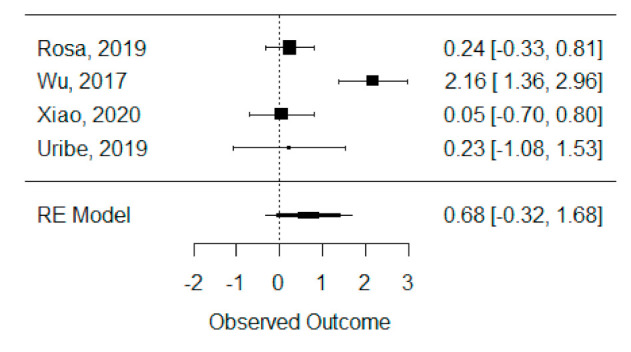
Forest plot for the standardized mean difference obtained for bone formation percentage analysis synthetic measures reported in the included systematic reviews.

**Figure 4 bioengineering-08-00076-f004:**
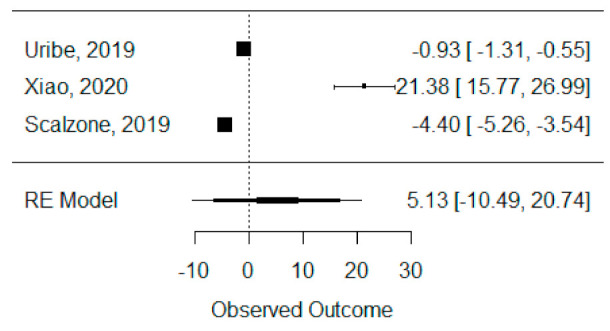
Forest plot for the standardized mean difference obtained for bone height synthetic measures reported in the included systematic reviews.

**Table 1 bioengineering-08-00076-t001:** Search keys in various databases.

Databases	Search Keys
PubMed	“Cleft Palate” [Mesh] OR “cleft Palate” OR “oral cleft *” OR “orofacial cleft *”. Filters: systematic reviews
WOS (all databases)	TS = (“cleft Palate” OR “oral cleft *” OR “orofacial cleft *”) AND TS = (“systematic review”)
Cochrane Library	#1 MeSH descriptor: [Cleft Palate] explode all trees #2 “oral cleft *” #3 “orofacial cleft *”
Scopus	“cleft Palate” OR “oral cleft *” OR “orofacial cleft *” AND “systematic review”
EMBASE	(‘cleft palate’/exp OR ‘cleft palate’ OR ‘oral cleft *’ OR ‘orofacial cleft*’) AND ‘systematic review’

**Table 2 bioengineering-08-00076-t002:** The characteristics of the included systematic reviews.

Author/Year	Design	Registration	No. of Trials and Design	ROB Tool	Quality of Evidence	Age of Participants	Intervention	Comparison Unit	Primary Outcome	Results
Da Rosa et al., 2019 [46]	SR/MA	Pros	RCT (4) RS (5) PS (1)	Cochrane guidelines	LROB (10)	7–16.4 years	rhBMP-2	Iliac crest bone graft	-Bone formation volume analysis -Bone filling percentage analysis	**(5 RCT)** Total (95% CI): BMP (54); ICBG (54); Weight 100% Std. Mean Difference 0.07 [−0.41, 0.56]; Heterogeneity: Tau^2^ = 0.14; Chi^2^ = 10.81, df = 8 (*p* = 0.21); I^2^ = 26%; Test overall effect: Z = 0.30 (*p* = 0.77). **(5 RCT)** Total (95% CI): BMP (95); ICBG (80); Weight 100% Std.Mean Difference 0.24 [−0.32, 0.80]; Heterogeneity: Tau^2^ = 0.38; Chi^2^ = 18.86, df = 7 (*p* = 0.009); I^2^ = 63%; Test overall effect: Z = 0.83 (*p* = 0.41).
Uribe et al., 2019 [47]	SR/MA	Pros	RCT (4) N-RCT (1)	Cochrane RBAT	Low-quality evidence (GRADE): HROB (5)	9.5–16.15 years	rhBMP-2 scaffold	Particulate iliac crest cancellous bone	-Bone filling	**(3 RCT)** Total (95% CI): BMP (16); Iliac crest (17) Weight 100% Std.Mean Difference −208.76 [−253.59, −163.93]; Heterogeneity: Chi^2^ = 0.38, df = 2 (*p* = 0.83); I^2^ = 0%; Test overall effect: Z = 9.13 (*p* < 0.00001).
Wu et al., 2017 [45]	SR/MA	Pros	RCT (14) N-RCT (26)	Oxford Centre for Evidence-Based Medicine 2011 Levels of Evidence	Best Evidence Synthesis Moderate Methodological Quality (7) Low Methodological Quality (18)	NR	1. Bone substitute materials (Bioglass, β-TCP, HA, Osteoblasts, BMP-2) 2. Supplementary materials (PRP, DDM, DBM, ADM, Bio-guide, Membrane, Periosteum, DBB) + ICBG 3. Autogenous bone grafts (cranium, rib, tibia and mandible grafts)	Iliac crest bone graft	-Bone filling rate -Clinical success rate	**BMP-2 (4RCT)** Total (95% CI): BMP (54); ICBG (46); Weight 100% Std.Mean Difference −2.16 [−10.10, 5.78]; Heterogeneity: Tau^2^ = 34.44; Chi^2^ = 7.84, df = 3 (*p* = 0.05); I^2^ = 62%; Test overall effect: Z = 0.53 (*p* = 0.59). **ADM + ICB (4RCT)** Total (95% CI): ADM + ICB (121); ICBG (171); Weight 100% Std.Mean Difference 1.34 [1.15, 1.55] Total events: ADM + ICB (98); ICBG (104); Heterogeneity: Chi^2^ = 4.46, df = 3 (*p* = 0.22); I^2^ = 33%; Test overall effect: Z = 3.80 (*p* = 0.0001). **Cranium Grafts** Total (95% CI): Cranium grafts (117); ICBG (193); Weight 100% Std.Mean Difference 0.86 [0.76, 0.97] Total events: Cranium grafts (92); ICBG (173); Heterogeneity: Chi^2^ = 4.18, df = 3 (*p* = 0.24); I^2^ = 28%; Test overall effect: Z = 2.53 (*p* = 0.01). **Rib Grafts** Total (95% CI): Rib grafts (38); ICBG (135); Weight 100% Std.Mean Difference 0.55 [0.37, 0.83] Total events: Rib grafts (15); ICBG (97) Heterogeneity: Chi^2^ = 3.52, df = 1 (*p* = 0.06); I^2^ = 72%; Test overall effect: Z = 2.87 (*p* = 0.004).
Khojasteh et al., 2015 [16]	SR	NR	CS (1) CR (3) CT (14)	Quality assessment (NR Tool)	Quality of Evidence-NR	NR	Cell group Osteoblasts (maxilla) MSC (Bone narrow) Growth factor rhBMP-2 Platelet rich plasma (PRP) Platelet rich fibrin (PRF)	Iliac Bone Graft Autogenous bone graft	-Bone formation	**Descriptive evaluation** **Cell group** Application of stem cells in alveolar cleft patients resulted in less than 50% of new bone formation, except in one case report, which was remarkable for 79.1% bone formation. MSCs seeded on DBM with calcium sulfate achieved 34.5% of new BF. **Growth factor** Application of BMP-2 with collagen led to 71.7% new BF. In one study, 90.9% of fistula closure was reported in cleft patients after the use of PRGF. PRP showed 71.27% BF compared to 47.47% in the control group, with 26.5% of secondary bone loss compared to 35.5% in the control group.
van Hout et al., 2011 [41]	SR	NR	RCT (2) CRR (1)	-	-	Children + Adolescents	BMP-2	Autologous bone graft	Bone Formation	**Descriptive evaluation** Dickinson et al. reported 95% of bone formation in the rhBMP-2 group compared to the 63% control group. Alonso et al. and Herford et al. reported less bone formation in the rhBMP-2 group compared to the control group (5.8% and 7%).
Xiao et al., 2020 [42]	SR/MA	NR	Case-control(9)	Newcastle-Ottawa scale	NOS 7 (5) NOS 6 (4)	NR	BMP-2	ICBG	-Bone graft filling rate -Volume of the bone graft area -Height of bone graft area -Bone graft density -Failure rate of bone graft -Infection after bone graft -Rate of oronasal fistula -Operative time length of hospital stay	**Bone graft filling rate** Total (95% CI): BMP-2 (54); ICBG (46); Weight 100% Std.Mean Difference −0.05 [−0.79, 0.69] Heterogeneity: Tau^2^ = 0.30; Chi^2^ = 6.79; df = 3 (*p* = 0.08); I^2^ = 56%; Test overall effect: Z = 0.13 (*p* = 0.90). **Bone graft volume** Total (95% CI): BMP-2 (24); ICBG (16); Weight 100% Std.Mean Difference −0.42 [−1.44, 0.60]; Heterogeneity: Tau^2^ = 0.41; Chi^2^ = 4.02; df = 2 (*p* = 0.13); I^2^ = 50%; Test overall effect: Z = 0.8 (*p* = 0.42). **Bone graft height** Total (95% CI): BMP-2 (14); ICBG (14); Weight 100% Std.Mean Difference −21.38 [−23.00, −19.76]; Heterogeneity: Chi^2^ = 4.88, df = 1 (*p* = 0.03); I^2^ = 80%; Test overall effect: Z = 25.81 (*p* < 0.00001). **Bone graft density** Total (95% CI): BMP-2 (37); ICBG (38); Weight 100% Std.Mean Difference −0.43 [−0.79, 1.64]; Heterogeneity: Tau^2^ = 0.56; Chi2 = 3.31; df = 1 (*p* = 0.07); I^2^ = 70%; Test overall effect: Z = 0.69 (*p* = 0.49). **Failure Rate** Total (95% CI): BMP-2 (316); ICBG (320); Weight 100% Std.Mean Difference 0.02 [−0.03, 0.06]; Total events: BMP-2 (316); ICBG (320); Heterogeneity: Tau^2^ = 0.00; Chi2 = 1.91; df = 4 (*p* = 0.75); I^2^ = 0%; Test overall effect: Z = 0.67 (*p* = 0.50). **Infection after bone graft** Total (95% CI): BMP-2 (294); ICBG (262); Weight 100% Std.Mean Difference 0.20 [0.05, 0.73]; Total events: BMP-2 (3); ICBG (11); Heterogeneity: Chi2 = 0.41; df = 1 (*p* = 0.52); I^2^ = 0%; Test overall effect: Z = 2.44 (*p* = 0.01). **Rate of oronasal fistula** Total (95% CI): BMP-2 (45); ICBG (31); Weight 100% Std.Mean Difference 0.41 [0.06, 2.63]; Total events: BMP-2 (1); ICBG (3); Heterogeneity: Chi2 = 1.15; df = 1 (*p* = 0.28); I^2^ = 13%; Test overall effect: Z = 0.94 (*p* = 0.35) **Operative time** Total (95% CI): BMP-2 (48); ICBG (34); Weight 100% Std.Mean Difference −3.64 [−7.35, 0.06]; Heterogeneity: Tau^2^ = 6.68; Chi2 = 15.06; df = 1 (*p* = 0.0001); I^2^ = 93%; Test overall effect: Z = 1.93 (*p* = 0.05) **length of hospital stay** Total (95% CI): BMP-2 (21); ICBG (27); Weight 100% Std.Mean Difference −1.97 [−2.41, −1.53]; Heterogeneity: Chi2 = 45.18, df = 1 (*p* < 0.00001); I^2^ = 98%; Test overall effect: Z = 8.74 (*p* < 0.00001).
Scalzone et al., 2019 [43]	SR	Pros	RCT (4)	Cochrane	GRADE-low High risk (2) Unclear risk (2)	>5 years old	rhBMP-2	ICBG	-Bone graft volume (6 months and 1 year) -Bone graft height (6 months and 1 year) -Length of hospital stay	**Bone graft Volume after 6 months** Difference means −14.410; Standard error 4.072; Variance 16.585; Lower limit −22.392; Upper limit −6.428; Z-value −3.538; *p*-value 0.000 MD −14.410; 95% CI −22.392 to −6.428; *p* = 0.000). **Bone graft Volume after 1 year** Difference means 6.227; Standard error 11.324; Variance 128.234; Lower limit −15.967; Upper limit −28.422; Z-value −0.550; *p*-value 0.582 (MD 6.227; 95% CI −15.967 to 28.422; *p* = 0.582). **Bone graft Volume after 1 year considering patient’s age** Standard Difference in means −0.493; Standard error 0.386; Variance 0.149; Lower limit −1.249; Upper limit −0.263; Z-value −1.278; *p*-value 0.201; (MD 30.000; 95% CI 11.593 to 48.407; *p* = 0.001).; Dickinson’s data (MD −0.493; 95% CI −1.249 to 0.263; *p* = 0.201). **Bone graft height 6 months** Difference means −18.737; Standard error 12.665; Variance 160.413; Lower limit −43.560; Upper limit 6.087; Z-value −1.479; *p*-value 0.139 (MD −18.737; 95% CI −43.560 to 6.087; *p* = 0.139). **Bone graft height 1 year** Difference means −4.401; Standard error 13.386; Variance 179.172; Lower limit −30.636; Upper limit 21.834; Z-value −0.329; *p*-value 0.742 (MD −4.401; 95% CI −30.636 to 21.834; *p* = 0.742). **Bone graft height after 1 year considering patient’s age** Standard Difference in means −6.523; Standard error 6.209; Variance 38.557; Lower limit −18.694; Upper limit 6.647; Z-value −1.051; *p*-value 0.293; (MD −6.523; 95% CI −18.694 to 5.647; *p* = 0.293). **Length of hospital stay** Standard Difference in means −1.146; Standard error 0.511; Variance 0.261; Lower limit −2.147; Upper limit −0.145; Z-value −2.244; *p*-value 0.025; (MD −1.146; 95% CI −2.147 to −0.145; *p* = 0.025).
Guo et al., 2011 [44]	SR	NR	RCT (2)	Cochrane	High risk (2)	Children and adolescents	rhBMP-2 iliac bone grafting + fibrin glue applied to the bone graft	ICBG	Complications, graft volume and grade of resorption, bone density and quality, alveolar ridge healing, nasal alar base augmentation, length of hospital stay, cost of surgery	**Descriptive analyses** **Traditional iliac bone graft versus artificial bone graft materials (+rhBMP-2)**. BMP-2 group (n = 9) had a score 0.9 point higher when compared to the iliac grafting group (n = 12) (mean difference (MD) −0.90; 95% confidence interval (CI) −1.16 to −0.64). After follow-up, the mean value of nasal alar base augmentation was 2.2 in the BMP-2 group (n = 9) compared with 2.0 in the iliac grafting group (n = 12), with no significance between the two groups (MD −0.20; 95% CI −0.41 to 0.01). **Traditional iliac bone graft versus traditional iliac bone graft plus fibrin glue** The average amount of graft resorption varied from 62.25% in the control group to 29.72% in the intervention group. The mean coronal bone volume was reported as 42.62 cm^3^ greater in the intervention group (64.32 cm^3^) when compared with the control group (21.70 cm^3^) (MD −42.62; 95% CI −64.25 to −20.99), and mean coronal bone density was 150.89 HU less in the control group (245.68 HU) than intervention group (396.57 HU) (MD −150.89; 95% CI −298.33 to −3.45). Regarding complications, dehiscence in the intervention group (infection in wound (RR 0.31; 95% CI 0.01 to 7.02); dehiscence (RR 2.79; 95% CI 0.33 to 23.52)).
Kamal et al., 2018 [24]	SR/MA	Pros	RCT (12) PS (10) RS(13)	MOOSE STROBE	Low risk (6) Moderate risk (9) High risk (11)	NR	Autogenous bone graft (iliac crest, tibia, mandibular symphysis, calvarium)	Growth factors Improved scaffolds and cell treatment Biocomposites and haemostatic agents	Reduction in postoperative volume of the cleft	**Reduction in postoperative volume using autogenous bone graft** Overall (Random effects): hedge’g SMD −1.91; Lower −2.25; Upper −1.57; *p*-value 0.000.; Heterogeneity: q-value =105.7; df = 24, *p*-value < 0.001; I^2^ = 77,3% (overall SMD = −1.91, 95% CI: −2.25 to −1.57, *p* < 0.001, I^2^ = 77.3%). **Reduction in postoperative volume using tissue-engineered bone substitutes** Overall (Random effects): hedge’g SMD −1.95; Lower −2.64; Upper −1.27; *p*-value 0.000.; Heterogeneity: q-value =28.8; df = 9, *p*-value 0.001; I^2^ = 68.7% (overall SMD = −1.95, 95% CI: −2.64 to −1.27, *p* < 0.001, I^2^ = 68.7%). **Subgroups analysis of studies using autogenous bone graft** Iliac crest- nr studies: 22; nr subjects: 371; hedges’g SMD (95% CI): −1.78 (−2.11 to −1.45); SE 0.169; Within group *p* value <0.001 Mandibular symphysial- nr studies: 1; nr subjects: 32; hedges’g SMD (95% CI): −3.12 (−3.95 to −2.28); SE 0.426; Within group *p* value < 0.001 Cranial- nr studies: 1; nr subjects: 10; hedges’g SMD (95% CI): −3.22 (−4.75 to −1.7); SE 0.777; Within group *p* value < 0.001 Tibial- nr studies: 1; nr subjects: 9; hedges’g SMD (95% CI): −2.46 (−3.74 to −1.18); SE 0.654; Within group *p* value < 0.001 *p* value between groups- 0.009 **Subgroups analysis of studies using tissue-engineered bone substitutes** Growth factors- nr studies: 6; nr subjects: 49; hedges’g SMD (95% CI): −2.34 (−3.39 to −1.28); SE 0.540; Within group *p* value < 0.001. Improved scaffolds and cell treatment- nr studies: 2; nr subjects: 11; hedges’g SMD (95% CI): −1.82 (−3.27 to −0.37); SE 0.742; Within group *p* value 0.014. Biocomposites and haemostatic agentes- nr studies: 2; nr subjects: 11; hedges’g SMD (95% CI): −1.20 (−2.35 to −0.05); SE 0.587; Within group *p* value 0.041; *p* value between groups 0.362.

ADM, acellular dermal matrix; df—degrees of freedom; BMP-2—bone morphogenetic protein–2; CI—confidence interval; CR—Case report; CS—Case series; CT—Clinical trial; DBB, deproteinized bovine bone; DBM, demineralized bone matrix; DDM, demineralized dentinal matrix; HA—hydroxyapatite; HROB—High risk of bias; HROB—high risk of bias; ICBG—Iliac cancellous bone graft; LROB–Low risk of bias; MD–mean difference; MSC—Mesenchymal stem cell; NOS—Newcastle-Ottawa scale; N-RCT—non- randomized controlled trial; NR—not registered; PRF—Platelet rich fibrin; Pros—Prospero; PRP—Platelet-rich plasma; PS—prospective study; *p*–*p-*value; RCR—Retrospective controlled review; RCT—randomized controlled trial; rhBMP-2—Recombinant human bone morphogenetic protein–2; RS—retrospective study; SMD–standardized mean difference; SR—systematic review; SR/MA, systematic review and meta-analysis; β-TCP—β-tricalcium phosphate.

**Table 3 bioengineering-08-00076-t003:** Quality assessment of the included reviews, using the AMSTAR2 tool.

Author/Year	PICO	Protocol	Inclusion Criteria	Comprehensive Search	Duplicate in Selection	Duplicate in Data Extraction	List of Excluded Studies	Description of Included Studies	Assessing Risk of Bias	Funding of Included Studies	Results of Statistical Combination	ROB Effect on the Statistical Combination	ROB in the Discussion	Discussion for the Heterogeneity	Publication Bias	Author’s Funding and COF Reporting	Overall Quality
Da Rosa et al., 2019 [46]	Yes	Yes	Yes	Partial Yes	Yes	Yes	Yes	Yes	Yes	No	Yes	No	Yes	No	No	Yes	Moderate
Uribe et al., 2019 [47]	Yes	yes	Yes	Partial Yes	Yes	Yes	Yes	Yes	Yes	No	Yes	Yes	No	Yes	No	Yes	Moderate
Wu et al., 2017 [45]	Yes	yes	Yes	Partial Yes	Yes	Yes	Yes	Yes	No	No	Yes	No	No	No	No	Yes	Low
Khojasteh et al., 2015 [16]	Yes	Partial Yes	No	No	No	No	No	Partial Yes	Partial Yes	No	No meta-analysis	No meta-analysis	No	No	No meta-analysis	No	Low
van Hout et al., 2011 [41]	No	No	Yes	Partial Yes	Yes	Yes	Partial Yes	No	Yes	Yes	No meta-analysis	No meta-analysis	No	Yes	No meta-analysis	Yes	Low
Xiao et al., 2020 [42]	Yes	Yes	No	Partial Yes	Yes	Yes	Partial Yes	Partial Yes	Yes	No	Yes	Yes	Yes	Yes	No	Yes	Moderate
Scalzone et al., 2019 [43]	Yes	Yes	No	Yes	Yes	Yes	Yes	Yes	Yes	Yes	Yes	Yes	Yes	Yes	Yes	Yes	High
Guo et al., 2011 [44]	Yes	Yes	No	Partial Yes	Yes	Yes	Yes	Yes	Partial Yes	Yes	No meta-analysis	No meta-analysis	Yes	Yes	No meta-analysis	Yes	Low
Kamal et al., 2018 [24]	Yes	Yes	Yes	Yes	Yes	Yes	Partial Yes	Partial Yes	Yes	No	Yes	No	Yes	No	No	Yes	Moderate

PICO—population, intervention, comparison, and outcome; ROB—risk of bias; COF—conflict of interests.

## Data Availability

The data presented in this study are available on request from the corresponding author.

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
