# Peer review of "Regenerative Strategies in Cleft Palate: An Umbrella Review"

_bioengineering, 2021, doi:10.3390/bioengineering8060076_

Round 1
Reviewer 1 Report
Review of the manuscript "Regenerative Strategies in Cleft palate: an umbrella review".
The manuscript touches an important topic, is well designed and well written.
Title and Abstract: Clear
Introduction: All important aspects addressed.
Materials and methods: All methodological details described.
results: clear
Discussion: methodological biases of the original and included studies could be more discussed so as to give input to the better design of the future studies.
Author Response
We thank the reviewer comments, which contribute to increase the manuscript quality.
The suggestion is very pertinent since the original studies and SR quality is fundamental to obtain quality evidence. We improved the discussion on this topic, as seen in lines 333-359: “Considering the risk of bias evaluated using the AMSTAR-2 tool, the items that presented more studies with low quality were: funding of included studies, discussion for the heterogeneity, ROB in the discussion, inclusion criteria and comprehensive search. In future systematic reviews, the authors should state their funding sources, namely by industry, as this may introduce a bias in the results presented. Most of the included studies failed to discuss the heterogeneity and the risk of bias, especially in justifying the inclusion of studies with different methodology and the consequent bias. Therefore, their conclusions must be interpreted with caution. Before starting the systematic review is essential to reduce methodological flaws, bias and duplication risk. Several SR included in this umbrella presented some gaps in the search description and inclusion criteria definition, which potentially increase the risk of publication and reduce the comprehensive nature of the review. Thus, the registration of the protocol is recommended and will enhance the robustness in further research. Although systematic reviews are considered the most reliable evidence, the studies included in each SR also had associated bias. The methodological heterogeneity includes differences in the trial settings, missing a priori adequate sample size calculation, type of sample included (e.g. type of cleft, age groups), intervention protocols, bone measurement tools and follow-up times. Other variables may affect the analysis of primary outcomes since they affect bone remodeling, namely, the position of teeth on bone graft, the cleft defect's width, and the volume of grafted bone. The primary outcomes may also be affected by the clinician´s expertise and the research group scientific proficiency. Secondarily, most selected studies were categorized as having a low or moderate overall quality, which may decrease the findings' certainty. Moreover, the included studies can perform an overestimation of the findings' effects due to the inclusion of several publications of a single study or by excluding studies in other languages. Finally, SR without meta-analysis may present a high risk of bias. Therefore, the findings of this umbrella should consider these limitations in the interpretation of results.”
Reviewer 2 Report
The review aims to evaluate the effectiveness of the current approaches in bone defects regeneration in non-syndromic patients with cleft palate. The review is well written, and concisely but exhaustively summarized the different methodologies currently adopted to treat bone defects.
I suggest the publication without any revision.
Author Response
We gratefully thank the reviewer comments.
Reviewer 3 Report
This is an interesting article and well-done study. However, there are some flaws those should be clarified.
Major concern: Authors analyzed the difference between ICBG and BMP-2 statistically and the difference between 2 techniques was the conclusion of this study. If authors just want to know the difference of these specific techniques, the title, introduction, and method should be revised per authors' purpose. However, the selection criteria for the review articles were too much non-specific and generalized. When looked at the selected review articles, all of them have BMP-associated research. When authors excluded 1297 articles by title and abstract, authors seemed to select BMP-associated research, intentionally. It is fine, but the title and method should be revised to the authors' intention.
Minor concern
1. In section 2.2. Some portion seemed to be missing. For example, "C—different available regenerative strategies;"
2. In section 2.3. Authors included "systematic reviews with or without ---". If it was said that "with or without", then it should be just said as "all systematic reviews".
Author Response
This is an interesting article and well-done study. However, there are some flaws those should be clarified.
Major concern: Authors analyzed the difference between ICBG and BMP-2 statistically and the difference between 2 techniques was the conclusion of this study. If authors just want to know the difference of these specific techniques, the title, introduction, and method should be revised per authors' purpose. However, the selection criteria for the review articles were too much non-specific and generalized. When looked at the selected review articles, all of them have BMP-associated research. When authors excluded 1297 articles by title and abstract, authors seemed to select BMP-associated research, intentionally. It is fine, but the title and method should be revised to the authors' intention.
R: We thank the reviewer comments, which contribute to increase the manuscript quality. This umbrella review aimed to compare all available regenerative strategies used in cleft palate patients, as stated in section 2.2: “Intervention—undergoing treatments approaches with regenerative strategies (All available treatment approaches for cleft palate closure: conventional autologous graft from different origins, alloplastic material, platelet-rich fibrin, platelet-rich plasma, resorbable collagen sponge, bovine-derived hydroxyapatite, allogeneic bone material, demineralized bone matrix, acellular dermal matrix and human bone morphogenetic protein 2”. So, we included all systematic reviews using any of these strategies that reported bone regeneration as the outcome in this umbrella review. However, most of the included systematic reviews focus on BMP and autologous bone grafts, as seen in table 2. This is justified because these two approaches are the ones more used in clinical practice. Nerveless, as seen in table 2, other treatments approaches are included, as substitute materials (Bioglass, β-TCP, HA), cells (MSCs, osteoblasts), membranes, PRP and PRF.
Minor concern
- In section 2.2. Some portion seemed to be missing. For example, "C—different available regenerative strategies;"
R: We appreciate the reviewer observation, and the manuscript was changed accordingly.
- In section 2.3. Authors included "systematic reviews with or without ---". If it was said that "with or without", then it should be just said as "all systematic reviews".
R: The manuscript was changed accordingly.
Reviewer 4 Report
This study has several strong points, namely the most complete and sophisticated study of Regenerative Strategies in Cleft palat.
In view of the rising interest in niche of "application of adult stem cell in cleft palat regeneration", this study will be a valuable reference for further analyses:
1. The identification of novel markers of quiescent adult neural stem cells (NSCs);
2. can estimate of the stiffness of the stem cell niche.
Author Response
We thank the reviewer comments, which contribute to increase the manuscript quality.
Although NSCs are not directly used for cleft palate regeneration, recent evidence suggests a link between cleft palate and cortical interneuronopathy. This way, these cells are important in clef patients.
Information regarding the link between clef palate and NSCs (lines 42-44), the biomarkers of quiescent adult stem cells (lines 78-80) and the importance of NSCs stimulation (74-80) was added to the manuscript.
Round 2
Reviewer 3 Report
The article has been revised successfully.
Author Response
Thank you for your commments.